# In Situ Changes in Mechanical Properties Based on Gas Saturation Inside Pressure Vessels

**DOI:** 10.3390/polym16091276

**Published:** 2024-05-02

**Authors:** Kwan Hoon Kim, Jae Hoo Kim, Dong Hwan Lim, Byung Chul Kwon, Jin Hong, Ho Sub Yoon, Sung Woon Cha

**Affiliations:** 1School of Mechanical Engineering, Yonsei University, 50 Yonsei-ro, Seodaemoon-gu, Seoul 03722, Republic of Korea; kimkevin99@yonsei.ac.kr (K.H.K.); imdh@yonsei.ac.kr (D.H.L.); swchiper@yonsei.ac.kr (B.C.K.); jin.hong@yonsei.ac.kr (J.H.); sorrytv@yonsei.ac.kr (H.S.Y.); 2Convergence Research Center for Solutions to Electromagnetic Interference in Future-Mobility, Korea Institute of Science and Technology (KIST), 5, Hwarang-ro 14-gil, Seongbuk-gu, Seoul 02792, Republic of Korea; murja@kist.re.kr

**Keywords:** polymer–gas mixture, batch process, magnetic sensor, polymer deflection, elastic modulus

## Abstract

In previous studies, difficulties were encountered in measuring changes within high-pressure vessels owing to limitations such as sensor connectors and sensor failures under high-pressure conditions. In addition, polymer–gas mixtures experience instantaneous gas desorption upon exiting high-pressure vessels owing to pressure differentials, leading to measurement errors. In this study, a device using magnetic sensors was developed to measure the real-time changes in gas-saturated polymers inside pressure vessels. Experiments on polymethyl methacrylate gas adsorption were conducted with parameters including pressure at 5 MPa and temperatures ranging from −20 to 40 °C for 60 and 180 min. It was observed that at −20 °C, the maximum magnetic field force density and deflection were 391.53 μT and 5.83 mm, respectively, whereas at 40 °C, deflection did not occur, with a value of 321.79 μT. Based on gas saturation experiments, a new model for deflection in high-pressure atmospheres is proposed. Additionally, an ANSYS analysis was conducted to predict the changes in Young’s modulus based on gas saturation. In previous studies, mechanical properties were measured outside the pressure vessel, resulting in an error due to a pressure difference, while the proposed method is characterized by the ability to directly measure polymer behavior according to gas saturation in high-pressure vessels using a magnetic sensor in real time. Therefore, it is possible to predict polymer behavior, making it easy to control variables in high-pressure polymer processes.

## 1. Introduction

Polymers are widely used materials in modern industries. Although polymer materials generally possess relatively inferior properties compared with metals and ceramics, they are the most commonly encountered materials in contemporary society owing to their cost-effectiveness and ease of processing. With increasing interest in these materials, significant progress has been made in the development and processing of polymer materials [1,2,3]. In the molding of resins, temperature is a critical factor, and variations in the processing temperature due to the type of polymer and additives are significant considerations in the development of important processing technologies. Changes in the state of polymers occur with variations in temperature. The two critical temperature points in this context are the glass transition temperature *T_g_* and the melting temperature *T_m_* [4]. The melting temperature refers to the temperature at which a solid substance changes its state to liquid. It is typically measured in crystalline polymers and signifies the temperature at which the crystallinity disappears. On the other hand, amorphous polymers soften over a broad temperature range referred to as the glass transition. In other words, amorphous polymers do not have a fixed melting temperature; rather, they exhibit a melting range because there is no specific point of solid-to-liquid transition in non-crystalline polymers [5,6].

The glass transition temperature is a critical property of amorphous polymers. Amorphous refers to a state in which the internal structure of a polymer lacks regularity or exhibits only minimally ordered arrangements in certain regions [7]. The glass transition temperature signifies the point at which the polymer matrix undergoes a transition, allowing relatively free movement above a certain temperature. Here, we refer to the temperature at which the formerly rigid glassy state transforms into a rubbery state akin to smooth rubber. In other words, the glass transition temperature denotes the temperature at which Brownian motion becomes significantly more vigorous compared to its previous state, transforming from the glassy phase to the rubbery phase [5,6].

Polymer materials, owing to their non-biodegradable nature, have been classified as major contributors to global environmental pollution, leading to ongoing efforts to reduce plastic usage [8]. Microcellular foaming represents a highly promising technology for reducing the use of polymer materials. Developed at Massachusetts Institute of Technology in the 1980s, this technology involves creating pores within polymers, each less than 10 µm in size, at a density exceeding 10^9^ pores/cm^3^ [9,10]. The fundamental principle of this process involves saturating polymers with a supercritical fluid under high pressure and utilizing thermodynamic instability to form pores. The resulting porous materials offer benefits including light weight, material savings, changes in properties such as decreased tensile strength and increased impact strength, and insulating and soundproofing effects [11,12,13].

A few experiments were conducted using a batch processing method for microcellular foaming. This process consists of two stages. In the first stage, the polymer is saturated with gas in a high-pressure vessel containing the gas. The subsequent process involves the formation of pores within the saturated polymer through the thermodynamic instability [14,15].

Polymethyl methacrylate (PMMA) is widely used in industry and research owing to its high transparency, weather resistance, and scratch resistance. Additionally, PMMA exhibits high carbon dioxide solubility compared to other polymers owing to its carbonyl group interactions [16,17]. Hence, research on PMMA and CO_2_ mixtures has garnered considerable interest.

When CO_2_ gas is saturated into polymers, it induces changes in the polymer matrix, leading to a decrease in the glass transition temperature *T_g_* [18,19,20]. However, Baldwin’s experiments failed to find a method to measure the glass transition temperature of polymers within high-pressure vessels [21]. Upon extracting the specimen from the pressure vessel, the gas begins to escape from the polymer, requiring a technique such as differential scanning calorimetry (DSC) to measure it. However, such devices for measuring the glass transition temperature cannot maintain high-pressure conditions, causing the gases saturated within the polymer to desorption, leading to experimental errors.

In response, Yoon and Cha [4] experimentally demonstrated that the glass transition temperature of polymer materials decreases proportionally with the saturation level of the gas using a device employing a magnetic switch. They proposed a new Cha–Yoon model that predicts the change in the glass transition temperature with respect to the gas saturation levels. However, they were unable to measure real-time changes through experiments, and the Cha–Yoon model had limitations, as it was not suitable for all polymer materials.

Additionally, experiments were conducted to measure the changes in the typical mechanical properties of polymers, such as impact and tensile strengths, using polymer–gas mixtures [22]. Through this process, it was observed that amorphous polyethylene terephthalate (APET) exhibited an increase in impact strength and volume with increasing gas saturation, whereas the tensile strength gradually decreased. However, gas loss occurred due to gas diffusion when the specimens were removed from the high-pressure vessels to measure their properties.

Brütting et al. [23] conducted experiments using polylactic acid (PLA) to investigate the relationship between the gas saturation level and glass transition temperature. The experiments utilized high-pressure DSC to measure the glass transition temperature, allowing for the maintenance of gas saturation levels during the experiment by injecting high-pressure gas. Experiments were also performed for various pressures and durations [24]. Through this experiment, the Chow model for the glass transition temperature and gas saturation level were validated for the first time [20]. The glass transition temperature of PLA decreased with increasing the carbon dioxide (CO_2_) content.

Previous research has extensively investigated the relationship between gas saturation and changes in the glass transition temperature; however, there has been limited focus on studies emphasizing the mechanical properties. In addition, previous studies have not directly measured the changes in properties within high-pressure vessels, and experiments have typically been conducted using only one material [23,25,26]. In this study, a device that can measure real-time changes in high-pressure vessels using a magnetic sensor was developed. The purpose of this study was to check the deformation of the polymer in real time due to gas saturation and temperature and to use it to check the change in the mechanical properties of the polymer.

## 2. Materials and Methods

### 2.1. Materials

In this study, PMMA with a purity of 99.98% (LX MMA, Yeosu City, Jeollanam-do, Republic of Korea) was used as the specimens. PMMA has a density of 1.19 and Young’s modulus of 2400 MPa. Additionally, the glass transition temperature was measured to be 110 °C, and the melting temperature was 158 °C. PMMA is a representative amorphous thermoplastic resin known for its high heat resistance and impact strength and is commonly used in various applications.

This study proposes a method for investigating the real-time changes in the mechanical properties of polymers owing to gas saturation within high-pressure vessels. PMMA, with its high gas saturation compared to other polymer materials, was chosen to measure the changes inside the pressure vessel. The specimens were designed using the developed device, with a thickness ranging from 1.3 to 1.4 mm, width of 90 mm, and height of 10 mm.

### 2.2. Blowing Agent

In batch foaming research, commonly used gases include nitrogen and CO_2_, which are selected according to the physical properties of the final blowing agent and the internal morphology requirements. The difference between these two solvents lies in the solubility and diffusivity of the gas, which leads to differences in the gas pressure [26,27]. Therefore, nitrogen is often used in physical foaming injection molding processes that require rapid solubility and pressure, whereas CO_2_, which is highly soluble, is preferred for batch foaming processes.

In this study, to investigate the *T_g_* values across various solubilities, it was crucial to use blowing agents with high solubility to enhance the experimental accuracy. In addition, depending on the type of gas, the diffusion rate and saturation amount differ. In other words, if a gas with low purity is used, errors due to various different gas molecules will occur [28,29]. Therefore, CO_2_ was selected as the optimal gas solvent. Specifically, CO_2_ (purity 99.9%, 40 L, Seoul Samheung Gastek, Seoul, Republic of Korea) was used for this study.

### 2.3. Experiment Setup

Changes in the glass transition temperature and mechanical properties occur depending on gas solubility. Therefore, the experiments were conducted using a batch processing method; the gas saturation process was investigated during batch processing. The radius and height of the pressure vessel were 100 and 200 mm, respectively, and the specimens and fixing devices were fabricated to match these dimensions.

In this study, a device was designed to measure the real time changes in the properties within high-pressure vessels. To enable the real-time measurements, a method was devised using magnetic sensors that could perform measurements outside a high-pressure vessel. A MAG3110 model from NXP (Eindhoven, The Netherlands) was used as the magnetic sensor. As shown in Figure 1, the polymer specimens experienced deflection owing to the solubility of the gas and weight of the magnet. Therefore, an appropriate fixture height was required, and the specimens had to be fixed to prevent changes in pressure during gas injection and discharge under high-pressure conditions.

To address this, a fixed jig was fabricated with a height of 40 mm and with a distance of 50 mm between the jigs, allowing for the fixation of the specimens inside the pressure vessel. The magnet was positioned at the midpoint between the jigs, 25 mm apart, resulting in a maximum distance of 40 mm between the magnetic sensor and the magnet. In addition, because magnetic sensors are highly sensitive to position, it was crucial to ensure that the positions of the magnet and magnetic sensor remained fixed. Therefore, a device was designed to fix the magnetic sensors in place and ensure accurate positioning.

The gas saturation experiments were influenced by temperature variations; therefore, the changes in temperature were measured using a temperature control device. For the high-temperature experiments, a band heater was used to control the temperature. For the ANSYS analysis, the mesh was set to 0.1 mm, and the parameter was set based on the physical property values of PMMA. In order to be the same as the actual experimental conditions, the boundary condition of the specimen was set. Both ends of the specimen were fixed, and a force of 15 g, which was the weight of the magnet, was applied to the position where the magnet was placed. For the low-temperature experiments, the pressure vessel was placed inside a cooling chamber capable of controlling the temperature down to −20 °C.

### 2.4. Characterization

To form polymer–gas mixtures, gas saturation was conducted using a batch processing device. The batch processing experiment involved six variables: the saturation pressure, saturation temperature, saturation time, pressure drop ratio, foaming temperature, and foaming time. However, this study only considered the gas saturation during batch processing; thus, variables related to foaming were not considered. Only the saturation pressure, temperature, and time were considered [30,31], and the gas saturation was measured using equipment (AR2130, OHAS Corp., Parsippany, NJ, USA).

The solubility of gas can be determined using Equation (1) as follows:(1)$$Gas\;sorption\left(\%\right)=\frac{{Weight}_{after}-{Weight}_{before}}{{Weight}_{before}}\times 100$$

In order to measure the elastic modulus, which is a representative mechanical property value, the deflection due to gas saturation was measured. In previous research, there were significant errors in property measurements because the specimens were removed from high-pressure vessels for measurement. In particular, the gas saturation in specimens removed from high-pressure vessels shows rapid changes owing to the pressure differentials. Therefore, in this study, the deflection was calculated using the magnetic sensor data measured based on the gas saturation, and the elastic modulus was determined using the ANSYS 2021 software.

Furthermore, experiments were conducted on seven specimens under each condition, and the average values of the results, excluding the maximum and minimum values, were obtained. The measurements were conducted on specimens where deflection due to actual gas saturation occurred, as shown in Figure 2. In addition, the experiment was conducted through temperature and pressure control to measure the deformation accordingly. The change in the magnetic field force according to pressure showed a change of ±2 to 3 µT and was ignored because it was within the standard deviation range. The change in the magnetic field force according to temperature did not change at all in the low-temperature experiment, and in the high-temperature experiment, large noise was generated when the band heater was operated. However, this was validated as it did not affect the tendency when the noise was removed.

## 3. Results

### 3.1. Magnetic Sensor Validation

Using the device depicted in Figure 1, we measured the changes in the magnetic field strength within the high-pressure vessel. As direct visual confirmation of the changes within the vessel was difficult, we observed the variations in the magnetic field strength caused by the magnets installed on the specimens. The PMMA specimens underwent changes in their internal matrix depending on the solubility of the gas, which in turn led to changes in their properties. Specifically, deflection occurred owing to changes in elasticity, which was measured using the magnetic sensors.

In the experiment, we verified the variation in magnetism based on the actual heights of the magnets within the pressure vessel. Measurements were conducted based on the height of the jig used in the experiment, which was 40 mm. At a height of 40 mm, the magnetic field strength was approximately 320 µT, and the standard deviation was 9.82.

The measurements of the magnetic field strength at different heights between the magnet and magnetic sensor yielded a graph similar to that shown in Figure 3, from which a fitting model was developed. Here, $M_{h}$ represents the magnetic intensity, and *h* denotes the height.
(2)$$M_{h}=1381\;\times \;exp(-0.037\;\times \;h)$$

Through the experiments, it was observed that as the height decreased, the magnetic field strength increased, as shown in Figure 3, confirming the relationship described in Equation (2). The discrepancy between the actual fitting model and experimental data was found to have a maximum error rate of 2.83%.

### 3.2. Gas Absorption

The gas saturation experiments in this study were conducted under various conditions, with the pressure fixed at 5 MPa and the temperatures set at −20, 0, 20, and 40 °C. The experiments were performed at time intervals of 30, 60, 120, and 180 min to assess the gas saturation trends in the PMMA over time. Figure 4 illustrates the variations in the gas saturation at different temperatures and durations. It is evident that the amount of CO_2_ dissolved in the PMMA increased steadily with time at all temperatures. However, as the temperature rose, the saturation level of the gas decreased. Notably, the highest saturation level was observed at −20 °C after 3 h, reaching 27.29%, whereas the lowest saturation level was recorded at 40 °C, amounting to 7.91%.

### 3.3. Measurement of Deflection Using Magnetic Sensor

This experiment aimed to assess the deformation of the PMMA specimens at different temperatures. The conditions were set with a fixed pressure of 5 MPa, whereas the temperatures were set at −20, 0, 20, and 40 °C. The experiments were conducted at time intervals of 60 and 180 min to examine the deformation of the PMMA over the saturation time. Real-time measurements of the changes in the magnetic field strength within the pressure vessel were conducted using the device shown in Figure 1.

Figure 5 illustrates the variation in the magnetic field strength with temperature. The figure indicates an increase in the magnetic field strength over time. The extent of the maximum change varied with temperature, with the highest magnetic field strength recorded at −20 °C (391.53 μT) and the lowest at 40 °C (321.79 μT). Additionally, magnetic field strengths of 384.96 μT and 367.46 μT were observed at 0 °C and 20 °C, respectively. These findings indicate that the variation in the magnetic field strength is influenced by the temperature, with changes of up to 21.67% being observed at different temperatures.

A change in the magnetic field force occurred owing to the deformation of the PMMA specimen. In other words, as the deflection increased, the distance between the magnet and sensor decreased, resulting in an increase in the magnetic field force. The magnetic field force data collected by the magnetic sensor were used to calculate the deflection of the specimen. This was achieved by calculating the deformation based on the temperature and time using Equation (1); the results are presented in Figure 6. When the maximum magnetic field force was used to measure the deflection, a deformation of 5.83 mm was observed at −20 °C, whereas almost no deformation was observed at 40 °C. Additionally, deformations of 5.47 mm and 4.31 mm were calculated at 0 °C and 20 °C, respectively.

To validate the previous experiments, the actual deformation of the specimens and the deformation calculated from the magnetic field force data were compared. The graph in Figure 7 compares the measured deflections of the actual specimens with those calculated from the magnetic data. As shown in Figure 7, the actual deflections were 5.83 mm at −20 °C, 5.25 mm at 0 °C, 4.50 mm at 20 °C, and 0 mm at 40 °C. When compared to the calculated data, the error rate was within ±4.2%, with a standard deviation of 0.37. Thus, it can be concluded that the gas saturation of a specimen can be determined using the maximum deformation.

Figure 8 depicts the specimens used in the actual magnetic field force experiments, showing the variations in deformation with temperature.

To determine the onset of deformation, the saturation time was further subdivided, and saturation experiments were conducted for 1 h. The onset point is identified by examining Figure 6. The onset point varied depending on the temperature. When saturating the gas at −20 °C, the deformation change began at 44 min, whereas at 0 °C, it began at 49 min. At 20 °C, deformation first occurred at 79 min. Deformation occurred more rapidly at lower temperatures, indicating faster gas diffusion. Additionally, no change was observed at 40 °C, suggesting that deformation does not occur without a certain level of gas saturation.

When comparing the saturation at the onset point, it was observed that at −20 °C, it was 15%, whereas at 0 °C, it was 14%. At 20 °C, deformation occurred at a saturation level of 11%. This indicates that even at low gas saturation levels, deformation can occur at high temperatures. This suggests that the glass transition temperature changed with the gas saturation, causing deformation despite the lower gas saturation reaching temperatures above the changed glass transition temperature. Therefore, it is possible to predict the change in the glass transition temperature based on the gas saturation.

### 3.4. Analysis of Elastic Modulus

To investigate the change in Young’s modulus due to the deformation, an analysis was conducted using ANSYS 2021. Figure 9 illustrates Young’s modulus as a function of the deformation obtained using ANSYS. Young’s modulus of the PMMA specimen in its neat state was 2400 MPa, and it decreased according to the gas saturation. At a Young’s modulus value of 1 MPa, a deformation of 108.044 mm occurred, whereas at 2400 MPa, the deformation was only 0.045 mm, indicating minimal change. Through graph analysis, a fitting model was developed, yielding the following equation:(3)$$E_{h}=108.04\;\times \;{\left(D_{h}\right)}^{-1}$$

In Equation (3), $E_{h}$ represents Young’s modulus at height *h* within the pressure vessel, and $D_{h}$ denotes the deformation at height *h*. Therefore, the relationship between Young’s modulus and the deformation is inversely proportional. In other words, as the deformation increased, Young’s modulus decreased.

Figure 10 presents the changes in Young’s modulus with time based on previously determined deflections for various temperatures calculated using Equation (3). For a duration of 3 h, Young’s modulus at −20 °C was the lowest at 18.672 MPa, whereas at 40 °C, it remained unchanged at 2400 MPa. Additionally, Young’s modulus at 0 °C was calculated to be 21.08 MPa, and it was 28.84 MPa at 20 °C. Consequently, it was observed that Young’s modulus exhibited significant variations with decreasing temperature.

Furthermore, Figure 10 illustrates the time points at which changes in Young’s modulus occurred. Considering the PMMA’s initial Young’s modulus of 2400 MPa, it is evident that the time points at which Young’s modulus changed varied depending on the temperature. At the lowest temperature of −20 °C, a decrease in Young’s modulus commenced at 43 min, which occurred more rapidly compared to higher temperatures. Similarly, at 0 °C, the change began at 49 min, and it started at 79 min for 20 °C. Interestingly, at the highest temperature of 40 °C, no change in Young’s modulus was observed. These time points correspond to the onset points observed in Figure 6.

Gas diffusion within the polymer structure saturated the polymer with gas, influencing the value of Young’s modulus of the intermolecular bonds and weakening the molecular chain structures. This weakening effect led to a reduction in tensile strength and elongation at break. Additionally, gas solubility enhances molecular mobility and reduces the glass transition temperature. Hence, higher gas solubility significantly alters the polymer properties.

## 4. Conclusions

Existing studies have encountered difficulties in measuring gas aturated polymer properties because of sensor connector limitations and sensor malfunctions under high-pressure conditions. In this study, we developed a new device capable of the real-time measurement of changes in gas-saturated polymer properties within a high-pressure vessel using a magnetic sensor. Gas saturation within the high-pressure vessel was achieved by utilizing the gas pressure difference to form a PMMA-CO_2_ mixture. The PMMA-CO_2_ mixture exhibited a decrease in tensile strength and Young’s modulus with increasing gas saturation, leading to deformation induced by the weight of the magnet inside the high-pressure vessel. The deformation of the specimens increased with the gas saturation. Additionally, changes in gas saturation occurred with temperature variations, resulting in different levels of deformation. Specifically, as the temperature decreased, the gas saturation increased, leading to a greater magnitude of deformation. Through Ansys analysis, the change in Young’s modulus according to sagging was measured, and through this, the value of Young’s modulus of the specimens obtained through experiments was confirmed. In summary, a PMMA-CO_2_ mixture was formed by applying a batch saturation process to a solid polymer, and the change in the mechanical properties according to the saturation amount of CO_2_ gas could be measured in a high-pressure vessel in real time. The process proposed in this study can be applied to PMMA as well as various polymers. If the change in the mechanical properties of the polymer–gas can be confirmed in real time, the deformation in the specimen in the foam molding process can be controlled. This can lead to the development of a new process by inducing a deformation in the polymer process under high pressure.

## Figures and Tables

**Figure 1 polymers-16-01276-f001:**
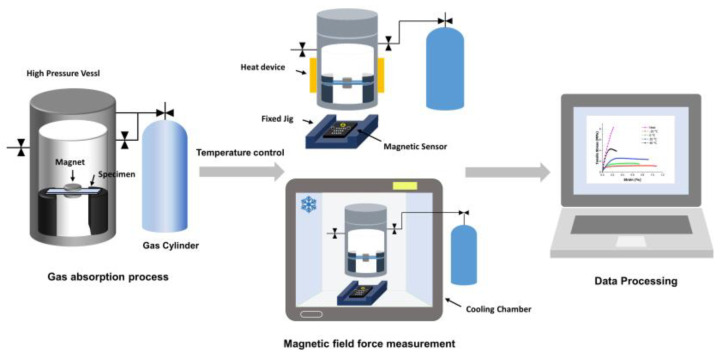
Schematic of in situ measurement of magnetic field force in a high-pressure vessel.

**Figure 2 polymers-16-01276-f002:**
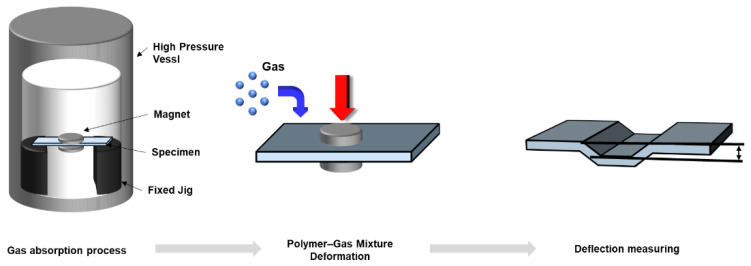
Schematic of measurement of deflection in high-pressure vessel.

**Figure 3 polymers-16-01276-f003:**
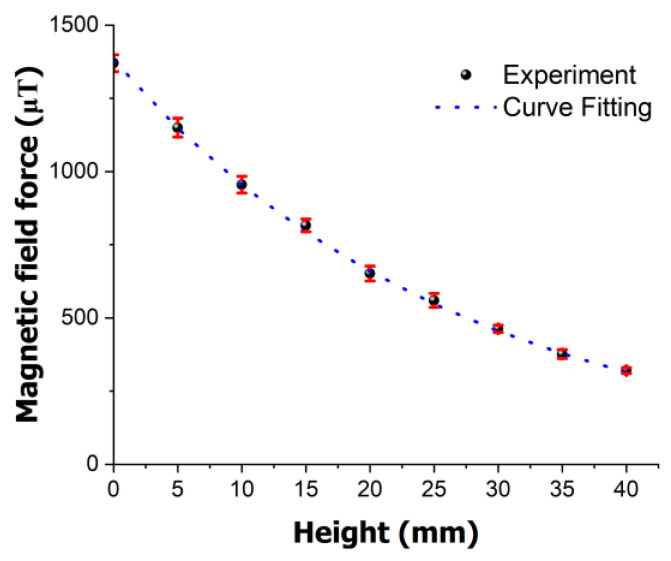
Variation in the magnetic field force with height between magnet and magnetic sensor in high-pressure vessel.

**Figure 4 polymers-16-01276-f004:**
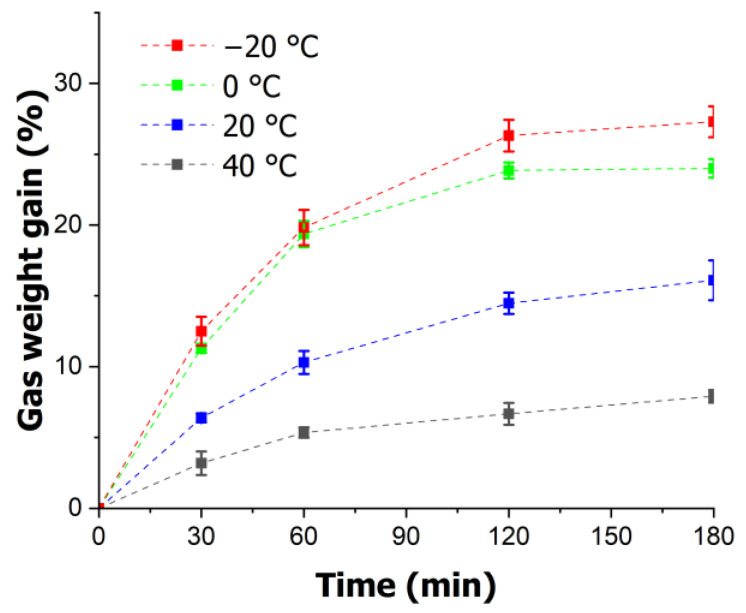
Temporal variation in gas saturation for various temperatures.

**Figure 5 polymers-16-01276-f005:**
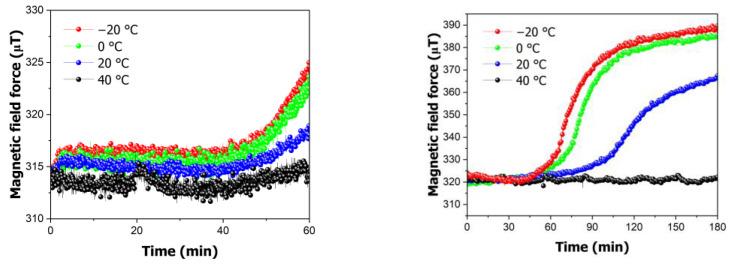
In situ variation in magnetic field force in high-pressure vessels at time intervals of 60 and 180 min.

**Figure 6 polymers-16-01276-f006:**
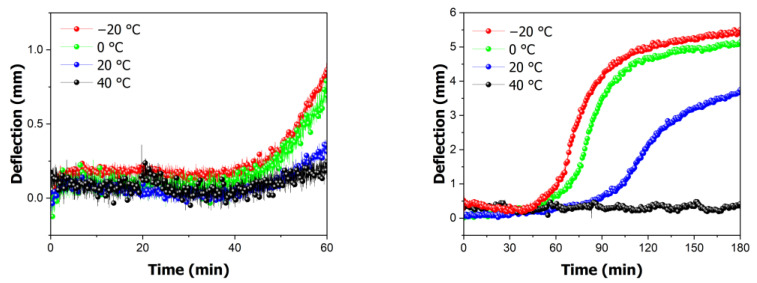
In situ variation in calculated deflection in high-pressure vessels at time intervals of 60 and 180 min.

**Figure 7 polymers-16-01276-f007:**
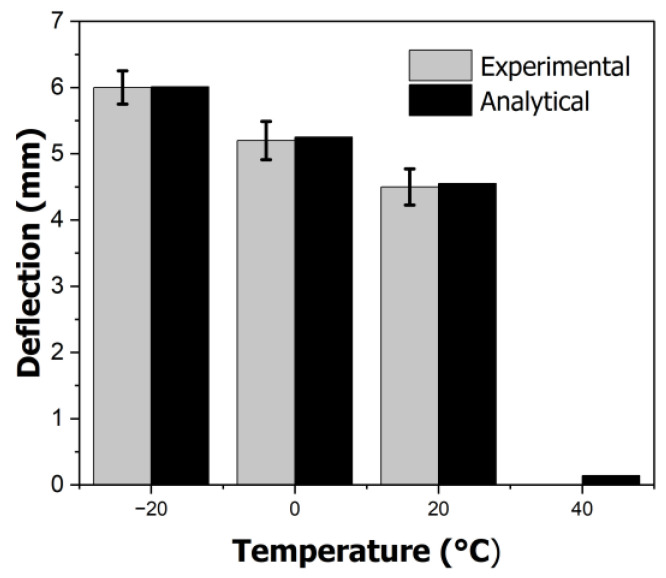
Comparison of experimental and analytical deflection of specimens.

**Figure 8 polymers-16-01276-f008:**
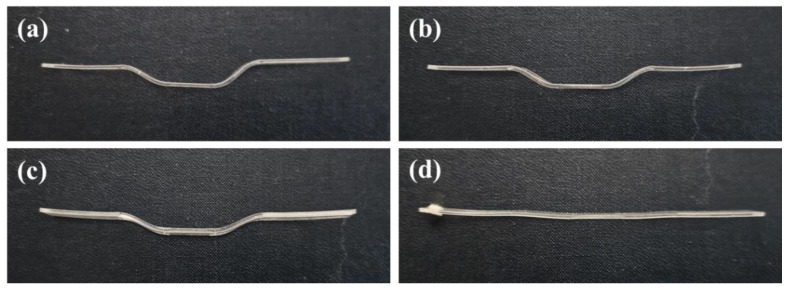
Specimen deformation with saturation temperature. (**a**) −20 °C, (**b**) 0 °C, (**c**) 20 °C, (**d**) 40 °C.

**Figure 9 polymers-16-01276-f009:**
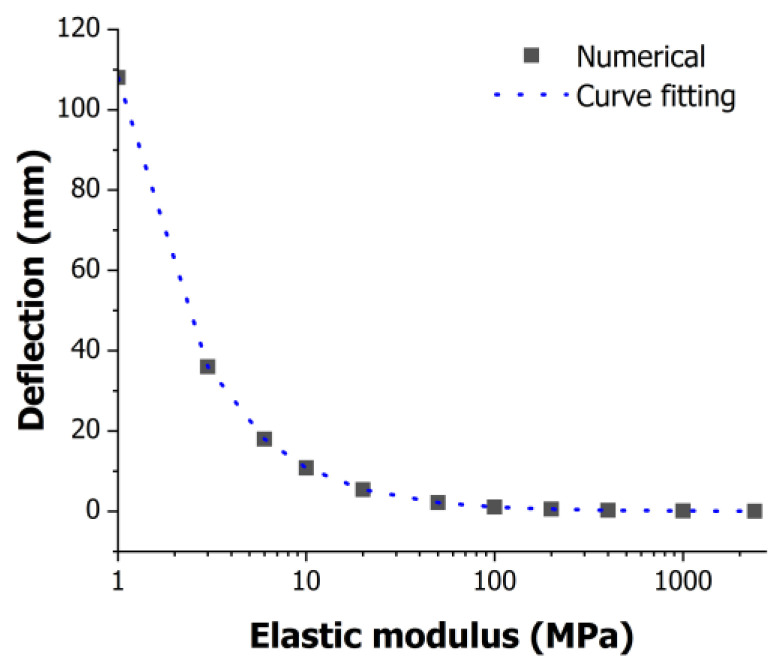
Relationship between deformation and elastic modulus using ANSYS analysis.

**Figure 10 polymers-16-01276-f010:**
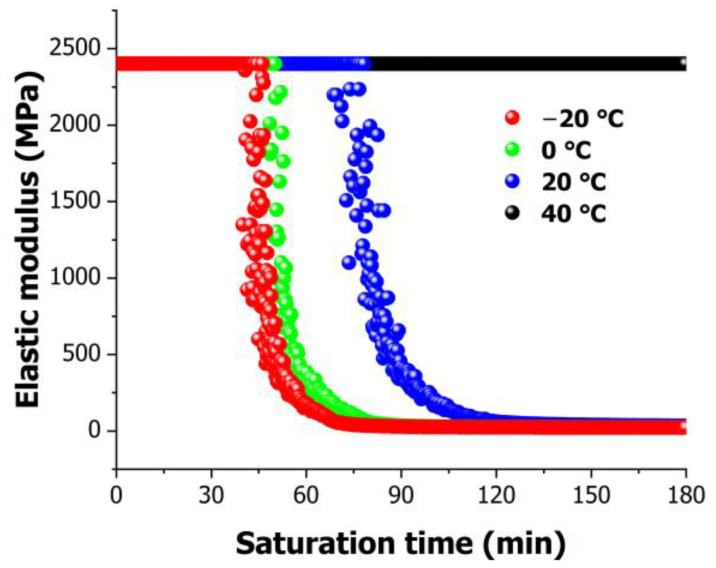
Temporal variation in elastic modulus for various temperatures.

## Data Availability

Data are contained within the article.

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
