# Peer review of "In Situ Changes in Mechanical Properties Based on Gas Saturation Inside Pressure Vessels"

_polymers, 2024, doi:10.3390/polym16091276_

Round 1

Reviewer 1 Report

Comments and Suggestions for Authors

In this study, a device using magnetic sensors was developed to measure the real-time changes in gassaturated polymers inside pressure vessels,and under the same pressure condition (5 MPa), adsorption experiments were conducted on gas-phase polymethyl methacrylate at varying temperatures ranging from -20°C to 40°C, with experimental durations of 60 minutes and 180 minutes. The proposed method enables the prediction of polymer behavior based on the gas saturation within high-pressure vessels, thereby facilitating the control of variables in polymer processing operations.The following are the reviewers' suggestions and we hope you will make appropriate changes and explanations. So that the readers can better understand and carry out the follow-up study.

1. Since this paper has developed a device utilizing magnetic sensors, it should, in its abstract, elucidate the novelty of the device, highlight how it differs from existing ones, and emphasize its prominent features.

2. In Chapter 1, the citations of references are not continuous, and variables such as temperature have not been italicized; please verify and make the necessary format changes according to the prescribed regulations.

3. In Section 2.2, provide a brief explanation as to why the selected CO2 concentration is 99.9%.

4. The content belonging to Section 2.3, as shown in Figure 1, is currently placed in Section 2.2; kindly rearrange it appropriately.

5. In Section 2.4, where it is mentioned that "the elastic modulus was determined using numerical simulation software, and experiments were conducted on seven specimens under various conditions", it is necessary to introduce the parameter settings within the numerical simulations and provide a concise description of the experimental conditions.

6. In Section 2.4, there is no mention regarding the consistency between the results obtained from numerical simulations and those derived from experimental findings.

7. In Section 3.1, the validation of the magnetic sensor did not take into account the variations in the magnetic field due to changes in temperature and pressure; please provide an explanation for this oversight.

8. The conclusion should revolve around organizing and summarizing the research content, and it would also be appropriate to consider incorporating potential application scenarios for the developed equipment in the concluding section.

9. Over the past five years, there is a limited number of referenced articles, amounting to only six. Additionally, there seem to be inconsistencies in the individual reference details; please cross-check and rectify them accordingly.

Reviewer 2 Report

Comments and Suggestions for Authors

The manuscript is on the influence of gas saturation on mechanical properties of polymethyl methacrylate (PMMA) in pressure vessels under conditions of polymer-gas mixtures. The research results are interesting and the manuscript is recommended for publication. A few points should be clarified before publishing.

1. Line 31. "industriese"?

2. It is worthwhile to more specifically define the purpose of the work, completing the “Introduction” section. The phrase “The purpose/objective of this research was” appears in various sections of the manuscript (lines 146, 181).

3. Fig. 5. The initial values of magnetic field force in the figures do not match, 315 and 320. Similar to Fig. 6.

4. References. The “References” section is poorly prepared. Correction required according to journal rules.

Round 2

Reviewer 1 Report

Comments and Suggestions for Authors

The author made modifications and improvements as required. However, there are the following issues: the image and text in Figure 1 are not clear. Formula 1 should be multiplied by 100% on the right side, which is more reasonable. The reference format is not standardized, such as carbon dioxide in references 25 and 26; In addition, references 9, 21, and others are not standardized, and there are also some references that do not have volumes; Please supplement and improve one by one.
